# Medical Liability: Review of a Whole Year of Judgments of the Civil Court of Rome

**DOI:** 10.3390/ijerph18116019

**Published:** 2021-06-03

**Authors:** Michele Treglia, Margherita Pallocci, Pierluigi Passalacqua, Jacopo Giammatteo, Lucilla De Luca, Silvestro Mauriello, Alberto Michele Cisterna, Luigi Tonino Marsella

**Affiliations:** 1Department of Biomedicine and Prevention, University of Rome “Tor Vergata”, 00133 Rome, Italy; michelemario@hotmail.it (M.T.); p.passalacqua92@gmail.com (P.P.); giammatteojacopo@gmail.com (J.G.); delucalucilla90@gmail.com (L.D.L.); mauriello.silvestro@gmail.com (S.M.); marsella.luigi@gmail.com (L.T.M.); 2XIII Section, Civil Court of Rome, 00192 Rome, Italy; alberto.cisterna@giustizia.it

**Keywords:** medical malpractice, medical liability, health law, civil tort litigation

## Abstract

Background: Complaints about medical malpractice have increased over time in Italy, as well as other countries around the world. This scenario, perceived by some as a “malpractice crisis”, is a subject of debate in health law and medical law. The costs arising from medical liability lawsuits weigh not only on individual professionals but also on the budgets of healthcare facilities, many of which in Italy are supported by public funds. A full understanding of the phenomenon of medical malpractice appears necessary in order to manage this spreading issue and possibly to reduce the health liability costs. Methods: The retrospective review concerned all the judgments drawn up by the Judges of the Civil Court of Rome, XIII Chamber (competent and specialized section for professional liability trials) published between January 2018 and February 2019. Results: The analysis of data concerning the involved parties showed that in 84.6% of the judgments taken into account, one or more health facilities were sued, while in 58.2% of cases, one or more health workers were present among the defendants. When healthcare providers are the only ones to be summoned, it is dentists and aesthetic doctors/plastic surgeons who undergo most of the claims. In the overall period analyzed, the amount paid was 23,489,254.08 EUR with an average of 163,119.82 EUR. Conclusion: The evidence provided by the reported data is a useful tool to understand medical malpractice in Italy, especially with regard to the occurrence of the phenomenon at a legal level, an aspect still hardly mentioned by existing literature.

## 1. Introduction

The concept of medical tort has ancient origins. In Roman law, if the Lex Cornelia settled a series of crimes by physicians, the Lex Aquilia (hence, the so-called “Aquilian” liability takes its etymological origin), introduced a penalty grading scale, including also further criminal hypotheses such as abandonment and experimentation as well as the possibility to compensate for the damage caused.

An article of the *Corpus Iuris Civilis* introduced the punishability of physicians for their unskillful conduct.

After the fall of the Roman Empire and the rise of the Goths, the issue was solved in a much more rough way. Visigoths asked physicians to pay a deposit before treating the patient, the Ostrogoths, quite simply, left doctors in the hands of patients’ relatives if a treatment-related fatal event occurred.

In the modern era, the general concept of professional malpractice appears in English legal doctrine since the early 17th century. In 1768, Sir William Blackstone, in his famous work, *Commentaries on the Laws of England*, introduced the concept of *mala praxis* (hence the term *malpractice*, currently used), “Injuries … by the neglect or unskillful [sic] management of [a person’s] physician, surgeon, or apothecary … because it breaks the trust which the party had placed in his physician, and tends to the patient’s destruction” [1].

In light of the above, it is doubtless that from the ancient times until today, despite the deep changes that have occurred, the ambivalent feeling underlying the patient–doctor relationship has remained substantially and justifiably unchanged and, perhaps, the resentment resulting from the disappointment in the event of failure to heal seems to have also strengthened as a consequence of the enormous scientific and technical progress achieved in the medical field.

With regard to the patient–doctor relationship, in fact, the so-called “paternalistic” approach has changed, moving from the idea of physicians as hieratic entities, nearly endowed with magical powers, to healthcare services providers to turn to and from whom to expect an outcome.

In modern times, especially in the last 2 decades, medical liability has received increasing attention both in the Italian [2] and international [3,4] medico–legal context.

The increasing attention that forensic scientists devote to medical liability is encouraged not only by the captivating issues arising from it but also by the significant rise in litigations resulting from MedMal (Medical Malpractice). This contingency is widely perceived as a real “malpractice crisis” [5].

An idea of the economic relevance of the issue comes from the reading of the report for the years 2016–2017 drafted by the Association of Italian Insurance (ANIA), which states that the amount relating to the premiums paid by public healthcare facilities amounted to 343.5 million EUR, compared to just over 87 million paid by private facilities. On the contrary, the amount of the insurance premiums paid by every single healthcare professional was ~208 million EUR. It was also reported that in 2016, insurance companies received approximately 14,803 claims, 6,884 of which were from healthcare facilities and 7919 from every single healthcare professional. The average cost per claim with regard to facilities was ~70,000 EUR, whereas for each professional involved it was ~40,000 EUR [6].

This situation, in less than a decade, has repeatedly required the intervention of the Legislator on a subject that, until the entry into force of Balduzzi Law (Legislative Decree no. 158/2012), had never been evenly discussed, since the case-law had been charged to draw up its features.

Over time, the development of a fruitful legal debate has also led judges to change their way of thinking and rethinking modern medicine, moving from the idea of medicine as an intellectual activity to that of a medical act performed by a healthcare provider.

All the above has led to considerable concern in the medical professional community, whose members, being involved in expensive trials with unfavorable rules of evidence, have significantly changed their way of thinking and experiencing their medical profession, being caught in a vicious circle resulting from a widespread mistrust towards patients, on one hand, and the rise of the well-known phenomenon of defensive medicine, on the other.

This debate culminated with the entry into force of Law 24/2017: “*Provisions on patient care safety and professional liability of healthcare providers*”, the so-called “Gelli-Bianco” Law.

One of the key targets of the aforementioned law was to standardize and align medical liability both in the civil and criminal context. In particular, with regard to the civil framework, the dividing line between the contractual liability by the healthcare facility, on the one side (Article 1218 of the Italian Civil Code—Liability of debtor: “The debtor who does not exactly render due performance is liable for damages unless he proves that the non-performance or delay was due to impossibility of performance for a cause not imputable to him”.), and the tortious liability by the individual professional, on the other (Article 2043 of the Italian Civil Code—Compensation for unlawful acts: “Any intentional or negligent act that causes an unjustified injury to another obliges the person who has committed the act to pay damages”.), has been definitively set out by Law 24/2017, a distinction having significant effects both on the burden of proof and on the limitation period. In Italy, in fact, in the civil field, the patient who believes to have suffered damage due to medical responsibility can request compensation directly from the healthcare professional or the healthcare facility, public or private, where the same operate. This setting, from a strictly legal point of view, could appear more favorable to the physician involved than to the healthcare facility; in the area of contractual liability (to which, in the light of Law 24/2017, the healthcare facility is subject) the principle of the presumption of fault applies, with the creditor (the damaged patient) only having the burden of proof of non-performance and the extent of the damage, while, conversely, the debtor (healthcare facility) will have to demonstrate the supervening impossibility of the performance for reasons not attributable to him in order to escape the obligation to pay compensation. On the other hand, if only the healthcare professional is called upon, the field is, except for particular cases, that of tort liability, in which the damaged party must prove all the constituent elements of the illicit act and, therefore, both the damage and the violation [7].

Despite the social and also economic relevance of the issue relating to litigations arising from medical liability, there is currently no public body charged to gather and examine data resulting from this phenomenon. Additionally, the continued absence of a national shared table for the judicial compensation for permanent disability from dynamic-relational damage makes it even harder to identify the exact correspondence between damage and amount of compensation, forcing case-law, on the issue and the merits, to a continuous search for uniform parameters and criteria.

The difficulty in obtaining a precise estimate has already been raised by various authors who, while acknowledging the value of many projects aiming to this objective, have observed that the available data are still not homogeneous and not performing a complete and global picture of the Italian MedMal phenomenon. The source of the data analyzed and published so far, both at the national and international level, is mainly provided by the insurance area, a fact that entails a clear underestimation of the phenomenon. Indeed, in Italy, a considerable portion of claims is directly managed by the Healthcare Facilities, failing to disclose to the companies insuring the facilities themselves.

In this respect, the lack of a “control room” able to integrate the data gathered by insurance companies, brokers, and organizations for the protection of patients and courts does not currently allow a precise estimate of disputes arising from Italian medical malpractice.

The purpose of this paper is to report the legal data relating to the phenomenon of liability arising from Med Mal by the analysis of a whole year of judgments of the Civil Court of Rome, the main Court at the national level by the number of litigations, representing about 20% of all national disputes [8].

## 2. Materials and Methods

The retrospective review concerned all the judgments drawn up by the Judges of the Civil Court of Rome, XIII Chamber, published between January 2018 and February 2019. The XIII Chamber of the Civil Court of Rome is the competent and specialized section for professional liability trials, including the medical sector. The judgments were provided by the Court pursuant to an agreement signed by the same and “Tor Vergata University of Rome”. With the exception of some replicated judgments, the documents were saved in PDF format and anonymized in order to preserve litigants’ personal identities and any connection between the tort in question and specific individuals or institutions. After the gathering and anonymization phase, 290 documents were analyzed. The actual analysis of the judgments was performed by three different auditors, experts in the field of medical liability. For the analysis, a working grid was prepared using the EXCEL program (office 365) in order to systematize the data mining. Furthermore, to reduce the risk of human error arising from the inter-subject variability between the three auditors, some locked fields with a drop-down list were added to the grid, so as to exclude the risk of inserting different definitions in overlapping fields. The grid is composed of the following items:-Judgment no.;-Occurrence year: the year when the event filing the lawsuit occurred;-Publication year of the judgment;-Difference between the registration and publication: a field to be filled with the difference, in terms of number of years, between the year of registration on the docket and the publication year of the judgment;-Difference between the event and claim: field to be filled with the difference, in terms of number of years, between the year of the tort bringing the action (if available in the text of the judgment) and the year of registration on the docket;-Medical specialty involved: a locked field has been set for this item with a drop-down list including all medical specialties acknowledged by Italian law;-Defendants/facilities: a locked field with a drop-down list to enter the type of facilities involved in the proceedings, if any, (available options: public health facility, private health facility, more public facilities, more private facilities, public and private facilities, no facility mentioned);-Defendants/persons: a locked field with a drop-down list to enter the type of professional(s) involved, should natural persons have been involved in the proceedings (available options: individual physician, several physicians together, non-physicians practicing a health profession, physician and non-physician health professionals, persons left unmentioned);-Provision of a court-appointed expert (CTU in Italy): a field with YES/NO locked options;-Inclusion of the CTU in the judgment: a locked field with a drop-down list to enter whether the provided technical report has or has not been fully or in part admitted by the judge (available options: yes/no/partially);-Civil conviction: a locked field relating to the outcome of the judgment with the possibility to select “yes” (if the defendant is guilty) or “no” (if the defendant is innocent);-Claimed damage: a locked field with a drop-down list to enter the type of damage claimed by the plaintiff (available options: injuries, death, consent, injuries and consent, death and consent);-Damage from consent: a locked field with a drop-down list to enter a possible dispute on the consent during the procedure and whether this dispute was deemed worthy of compensation by the judge (available options: claimed and paid, claimed and not paid, not claimed);-Damage for loss of chance: a locked field with a drop-down list to enter whether a damage from loss of chance was claimed or otherwise paid by a plaintiff during the proceeding (available options: claimed and paid, claimed and not paid, not claimed and paid, not claimed);-Compensation paid out;-Compensation claimed;-Compensation for consent: a field to enter the amount of compensation for damage from consent specifically if liquidated by the judge and mentioned in the judgment.

To complete the filling of the grid, the data were analyzed and checked by a fourth auditor different from those who dealt with the previous data mining from the judgments in order to exclude any inconsistencies and errors which, if any, would have required a re-evaluation of the judgments. During the analysis, it was found that 10 judgments did not deal with cases concerning medical professional liability lawsuits, but they were rather related to motor T.P.L. (RCA in Italy) and veterinary professional liability. It is for this reason that the latter judgments were not taken into account during the data mining phase.

Finally, it should be noted that during the data mining it was not possible to fill in every single field per judgment, since the judgments were drafted in an open form that might not contain all the same information. The next processing phase was carried out through a data-cross match using the EXCEL program filter function (Windows Office 365).

## 3. Results

### 3.1. The Timeline

With regard to the proceeding timeline, analysis of the data showed an average period of 5.3 years (DS ± 3.87) between the claimed event and the start of the actual dispute, while the interval between the registration on the docket (that is its start) and the publication of the judgment was about 4.3 years (DS ± 1.84).

However, it should be considered that in a few cases analyzed, a previous proceeding had already been filed and settled; in eight judgments the argument referred to a previous criminal conviction and in another three judgments an appeal was previously filed pursuant to art. 696 bis. Nevertheless, the final data concerning the proceedings timeline are to be considered reliable by virtue of the numerical scarcity of the aforementioned occurrences. Most of the judgments analyzed concerned proceedings registered from 2010 to 2017; of these, approximately 54% had started between 2013 and 2015.

### 3.2. The Parties

The analysis of data concerning the involved parties showed that in 84.6% of the judgments taken into account, one or more health facilities were sued, while in 58.2% of cases, one or more health workers were present among the defendants. On the 43 judgments (15.4% of the total) in which only physicians were involved, 25 (or 58.1%) concerned dentists, while a smaller part (six judgments) were about cases related to plastic surgery, three orthopedics, two ophthalmology, and another two oncology, all different in the remaining five. It is important to highlight that in 74% of the judgments concerning dentistry malpractice, dentists were singly summoned before civil courts, whereas aesthetic surgeons/doctors were singly summoned in 27% of cases.

### 3.3. Judgment Outcome

In 51% of cases (144 judgments out of 280 analyzed), some medical malpractice profiles were detected; in 46% of cases (128 judgments), liability was excluded, whereas in 3% (eight judgments) no judgment was settled, as a result of the reaching of a settlement agreement or the ineligibility of the claim itself (Figure 1).

In 280 judgments analyzed, the judge provided the execution of a court expert’s report in 93% of cases (260 lawsuits); in the remaining 7% (20 judgments) any type of technical investigation was required by the judge. In these cases, proceedings concerned either the ineligibility of the claim or a previous appeal pursuant to Article 696 bis of the Italian Code of Civile Procedure or the case could be solved without the need of a court expert. The acknowledgment by the judges of the court-appointed experts’ outcomes occurred in 92.7% of cases (241 judgments out of 260 in which an expert witness was appointed by the Court). A partial acknowledgment occurred in 3.5% of cases, while in 3.8% the conclusions by the court-appointed experts were entirely rejected by judges.

### 3.4. The Damage

With reference to the data relating to the type of the most claimed damage, it is clear that physical injuries were claimed by the injured parties in 80.6% of cases. Deaths were significantly less and affected only the remaining 19.4% of disputes. However, only in one case, with a conviction of the defendants, improper informed consent was complained as a damage (Figure 2).

During the analysis, the authors differentiated those cases claiming both injuries or death and the prejudice to the right to self-determination arising from an omitted or poor consent; the latter prejudice was explicitly claimed by the plaintiff in about 24.9% of cases. This prejudice, despite having been claimed in about one out of four cases, was admitted only in 42% of the cases in which it was complained, with an average compensation of 8648 EUR.

With regard to the damage claimed, the 278 judgments taken into consideration showed that in 22 of these, approximately 8%, the plaintiff requested to the judge damages from loss of chance. Furthermore, it should be noted that in two judgments such damage was liquidated despite the lack of an explicit claim by the party, or at least, the reading did not show a request to that effect.

### 3.5. Medical Specialties

For each medical field, the number of proceedings along with the judgments dealt with facilities liability and medical professionals’ malpractice were extrapolated. The data obtained were then compared as seen in Figure 3.

It should be noted that in 280 judgments analyzed, eight were left undefined (termination of the dispute subject-matter and reaching of a settlement agreement, etc.) and so counted as non-conviction.

As many as 172 out of 280 judgments involved only six medical specialties, thus representing 61.4% of the cases, with a total number of 107 convictions, or 74.3% of the total convictions. Table 1 shows the ratio of lawsuits and convictions for the six most involved specialties.

### 3.6. Compensation

In the overall period analyzed, the amount paid was 23,489,254.08 EUR (except for any legal interest or court fees) with an average of 163,119.82 EUR. The highest compensation paid for a judgment was 4,741,34.43 EUR and concerned a neonatology lawsuit. Table 2 shows the average compensation paid for each of the six most involved specialties.

In 117 out of 280 judgments, it was also possible to extrapolate the data relating to the average sum of compensation claimed, amounting to 473,694.63 EUR, or about three times the average value of the compensation actually paid.

## 4. Discussion

As reported in the opening section, the overall analysis of the data relating to researches carried out on medical disputes shows a clearly growing trend, so that in some states, legislators have provided regulatory measures, reducing its excessive increase [9], especially in case of disputes initiated in the absence of any liability or of any sort of damage occurrence.

The importance and relevance of the phenomenon are suffered by the civil society too, with an impact at an economic and social level.

With regard to the economic framework, in the United States, a recent estimate of the annual costs of this phenomenon showed that they amount to 55.6 billion USD, or 2.4% of total healthcare spending [10].

In Italy, as mentioned in the statistical bulletin published by the Insurance Authority (IVASS) [11], the insurance premiums collected from civil liability for medical malpractice-related risks (both for healthcare facilities and individual professionals) amounted to 612 million EUR in 2018, up compared to 2017 (+3.7%).

The average premium covering a public healthcare facility costs was 456.000 EUR (+25.5% compared to 2017). The value is 24 times higher than that for a private healthcare facility.

The economic burdens weighing on public facilities are not only represented by insurance premiums, but also, of course, by compensation paid using the facilities’ own funds, being lower than the expected deductible or SIR (Self Insured Retention) pursuant to the insurance contract. It should also be noted that an increasing portion of Italian health facilities has opted to self-insure (self-risk retention), as an exclusive form of protection. The document drawn up by IVASS reported that the allocations made in 2017 amounted to 592.4 million EUR (+16% compared to the previous year) [11].

If, on one hand, the approach represented by self-risk retention seems to be potentially virtuous, on the other, it requires the implementation by the healthcare facility of adequate internal structures made up of healthcare professional experts in the field, able to settle claims quickly and ensure a proper defense during proceedings.

The economic burden weighs directly on every single healthcare provider, especially if he/she is a physician; the IVASS statistical bulletin shows that the average premium paid by medical providers is 1001 EUR, against the 183 EUR paid, on average, by non-medical health professionals.

As mentioned at the beginning of this section, medical malpractice costs are also social.

Numerous authors have identified a growing use by the medical community of the so-called “defensive medicine” as an indirect consequence arising from health litigations [12], which, however, do not lead to an improvement in quality of care and outcomes for patients [13].

Furthermore, it was highlighted that to be involved in a legal action may be stressful for the healthcare professional (also due to the length of the judicial process), who may suffer damage to his/her reputation and a loss of trust in the physicians’ own performance by other patients [14].

With respect to the proceeding timeline, the analysis carried out made it possible to highlight two important facts: on average, around 5.3 years elapse between the contested event and the start of the litigation, whilst the time elapsing between the registration of the proceedings (coinciding with the start of the same) and the publication of the judgment (coinciding with its end) is around 4.3 years. From the collected data it is clear that, in Italy, the proceedings relating to medical professional liability lawsuits are characterized by a long latency period between the damaging event and the reach of judgment by the injured party or his/her heirs. This discrepancy can be explained by patients’ delayed comprehension of the damage suffered as a result of both inadequate medical assistance and the need to find, under the Italian civil code, the liable persons and the “abstractly eligible” breaches occurred before reaching the judgment.

This period of time, certainly important, has a great impact on the economic management of the capital placed in reserve by health companies and insurance companies, as well as on the cost of the legal fees needed for the management of each proceeding.

It should also be considered that the analysis focused only on actual judicial actions. It is, therefore, very likely that in some cases, the possibility of an out-of-court resolution of the dispute was considered prior to the settlement of the judgment. n this regard, it should be noted that Article 8, Law 24/2017, provides that the submission of an appeal pursuant to Art 696 bis of the Italian Code of Civil Procedure or, alternatively, a mediation process attempt, is a prerequisite of admissibility for compensation.

The reported data confirm what has already been highlighted by various authors on the most involved specialties in lawsuits and convictions [15,16,17].

The reported analysis has shown that most of the disputes involved health facilities, especially public, compared to 15.4% in which health professionals were directly and exclusively summoned. Among the latter, physicians were the most involved (this has indirectly been confirmed by the significantly higher average cost of medical insurance policies than those intended for other health professionals). In this regard, it has been estimated that, for consultants specializing in specialties at higher risk, the possibility of being involved in litigation within the age of 45 is 88% [18].

By our analysis, it can be concluded that when healthcare providers are the only ones to be summoned, it is dentists and aesthetic doctors/plastic surgeons who undergo most of the claims (74%).

This figure was entirely expected, since the dentist and aesthetic doctor medical practice, mainly carried out on a private basis, exposes healthcare professionals themselves as a result of the signing with the patient of a specific “contract” under the ex-art. 1218 of the Italian Civil Code. It should also be noted that for these two specialties, Italian law aims to admit a substantial result obligation [19].

This investigation documented that in 51% of judgments, the defendant party was held liable. The comparison of this result with the data reported by ORME (Observatory on Medical Liability), referring to less than a decade ago, reveals that there is a difference arising from the decrease of data concerning the acknowledgment of liability against the defendants; in fact, ORME analysis revealed a liability rate exceeding 60% [20].

The data concerned are, in the authors’ opinion, extremely meaningful, and would require a regular updating in order to find out the impacts resulting from legislative changes, doctrine, and especially case-law, with their potential implications also in the practice, claims management, and insurance market focused on malpractice.

With regard to the type of damage claimed by patients, data suggest that most cases dealt with physical injuries, although compensation was requested by heirs for the patients’ death only in about 20% of cases.

In addition, in one out of four cases, injuries resulting from an infringement of the right to self-determination were also claimed, and this results, practically, in a lack of informed consent to the medical act.

The concept of informed consent, used with this meaning for the first time in 1957, has been recently introduced in Italian legislation with the Law n.219/2017, which is the first law on this issue. It can be defined as a real communication path between doctor and patient, by which the doctor makes the patient competent about all medical information needed in order to make him/her aware and able to voluntarily and consciously accept or refuse the planned medical treatment. The three fundamental criteria that are needed for informed consent are that the patient must be competent, adequately informed, and not coerced [21].

For years, the international scientific literature has shown that effective communication with the patient is an undeniable advantage for the doctor too, leading to higher patient compliance with the planned treatment as well as better tolerance in the event of complications [22].

In Italy, the importance of the informed consent concept in modern medical practice has been strengthened after the entry into force of Law 219/17, “Regulations in the area of informed consent and advance treatment directives” [23]. It is assumed that the introduction of this new law, by which new and clear directives have been set out imposing both on the doctor and health facility a huge attention in the collection and documentation of the consent, could lead the civil judge to an increase in interest with respect to this type of prejudice. Furthermore, a possible rise in new cases worthy of protection and therefore of compensation following the new regulatory provisions on the shared care plan should not be underestimated [23].

Given that, under the Italian Civil Law, the amount of compensation for damage arising from omitted or poor consent is equitably liquidated by the court, as set out in Article 1226 of the Italian Civil Code (Art. 1226 of the Italian Civil Code: “If damages cannot be proved in their exact amount, they are equitably liquidated by the court.”), it was observed that this prejudice, explicitly mentioned in a few judgments, was liquidated in a range fluctuating between 2000 EUR and 20,000 EUR, with an average of 8648 EUR. Given the variability arising from the economic assessment of injuries, it might be helpful to draw up a compensation grading scale allowing the various figures involved to predict, albeit roughly, the economic risk in the event of conviction, allowing an equitable and standardized, as much as possible, “monetization “of the prejudice.

Particularly worthy of mention are data concerning paid compensation for damage from loss of chance. This case is actually a relative novelty in the field of medical liability that is still not fully accepted and on the assessment of which there are still no shared guidelines either in the legal medical context or in the case-law, uncertainty that is also reflected on the verification of the causal link between conduct and event, on which there is not yet a universally shared orientation [24].

The analysis carried out has ascertained that the damage from loss of chance is still a reality hardly admitted by the court, having been confirmed only in 3% of the analyzed judgments; given the shortage of the sample collected, it might be useful, in a further analysis, to focus attention on the jurisprudential implications from this particular type of damage.

## 5. Conclusions

The evidence provided by the reported data are useful tools to understand medical malpractice in Italy, especially with regard to the occurrence of the phenomenon at a legal level, an aspect still hardly mentioned by literature.

Some of the findings, especially those concerning the most affected specialties and the type of damage, confirm what has been already outlined by several authors at an international level. Conversely, with regard to the case of loss of chance, damage arising from the breach of the right to self-determination and the amount of compensation, the analysis carried out has given rise to some new and interesting issues, with potential doctrinal and practical implications. In this sense, the information provided could be used not only by forensic scientists working in the field, but also by health facilities that, due to the growth of the MedMal phenomenon, have to deal with a considerable number of disputes with significant disbursements of money.

A new study carried out, taking into consideration a longer period, would allow researchers to obtain a larger sample and, consequently, higher statistical reliability. In addition, it might be interesting to consider in a future analysis the outcome of the judgments following the registration on the docket after the entry into force of Law 24/2017, in order to understand the effect of the recent regulatory intervention.

### Limitations of This Study

It should be noted that during the data mining it was not possible to fill in all fields for every single document analyzed, since the judgments, being drafted in an open form, might not contain the same amount of information. For example, in some cases, it was not possible to deduce from the judgments the exact claim by the plaintiff, while in many cases the economic quantum requested by the plaintiff was not mentioned. However, this is not believed to have significantly influenced the validity of the data collected. It could also be useful to obtain the entire case file for each judgment analyzed. Unfortunately, despite the introduction in Italy of the Telematic civil proceedings, actually, it is not possible to access Telematic civil trials.

## Figures and Tables

**Figure 1 ijerph-18-06019-f001:**
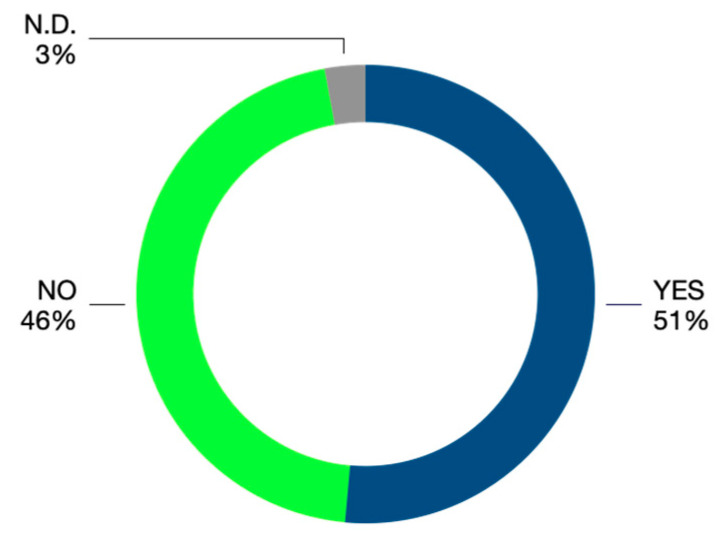
Judgment outcomes.

**Figure 2 ijerph-18-06019-f002:**
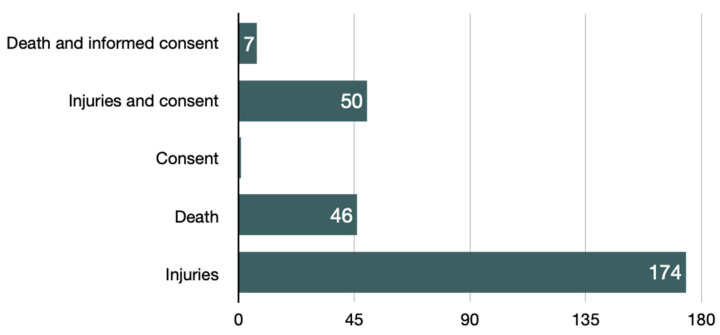
Categories of damages claimed by patients in court.

**Figure 3 ijerph-18-06019-f003:**
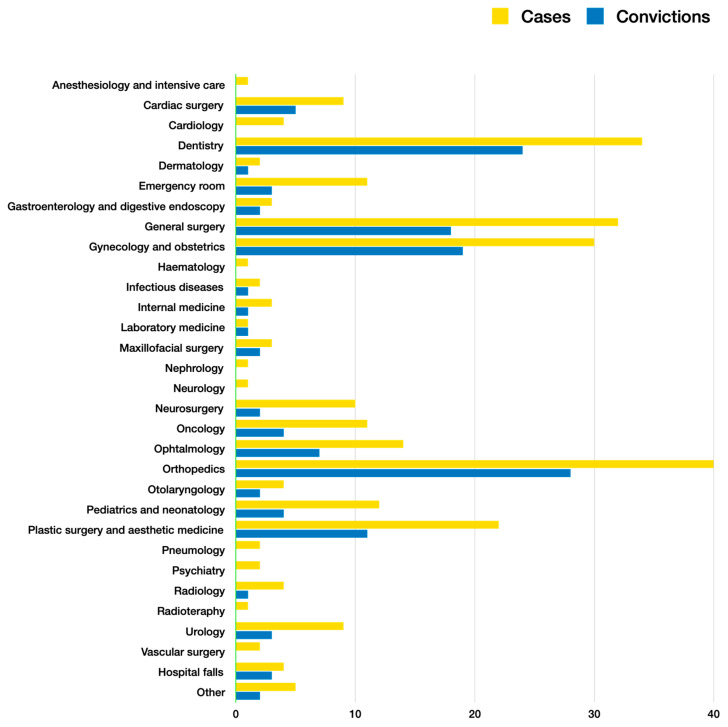
Cases and convictions divided by medical specialty. The ordinates show the branches involved and the abscissae show the number of total cases and the number of convictions.

**Table 1 ijerph-18-06019-t001:** Most involved specialties and risk of recognizing liability in court.

Type of Specialty	Cases	Convictions	Ratio Convictions/Cases
Dentistry	34	24	70.59%
Orthopedics	40	28	70%
Gynecology and obstetrics	30	19	63.33%
General surgery	32	18	56.25%
Ophthalmology	14	7	50%
Plastic surgery and aesthetic medicine	22	11	50%

**Table 2 ijerph-18-06019-t002:** Average compensation for the most involved specialties.

Medical Specialties	Average Compensation Admitted by Judgment
General surgery	205,579.00 EUR
Gynecology and obstetrics	105,343.00 EUR
Orthopedics	74,762.00 EUR
Ophthalmology	42,266.00 EUR
Plastic surgery and aesthetic medicine	20,775.00 EUR
Dentistry	19,096.00 EUR

## Data Availability

Data available on request due to privacy restrictions. The data presented in this study are available on request from the corresponding author.

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
