# Peer review of "Medical Liability: Review of a Whole Year of Judgments of the Civil Court of Rome"

_ijerph, 2021, doi:10.3390/ijerph18116019_

Round 1
Reviewer 1 Report
I have read with great interest the work entitled “Medical Liability: Review of a Whole Year of Judgements of the Civil Court of Rome". Overall, the work is well written and the experimentation well conducted. In order for it to be considered for publication, some insights are necessary.
- In the introduction it should be specified whether there are differences from a legal point of view between the cases in which the individual doctor is cited and those where an entire health facility is cited.
- Line 194 Authors should introduce the standard deviation beyond the mean.
- Line 232 6%+20%= 100,6%? Authors should report accurate data to one or two decimal places.
- Authors should use different colors in the figures. The shades used are close and make the figure not interpretable quickly.
- The authors collected data from a single center. There are reports that speak of medical malpractice for the period indicated also at national level (https://www.marsh.com/content/dam/marsh/Documents/PDF/it/it/Marsh_Medmal_report_2020_web.pdf). Authors should introduce a parallelism into the discussion.
Reviewer 2 Report
Medical Liability: Review of a Whole Year of Judgements of the Civil Court of Rome
Review- Comments
In the Introduction section the authors should provide a brief description of the procedure/s applied in cases of malpractice complaints in Italy, showing the role of the insurance companies, healtcare facilities and of the Civil Court.
Lines 92-96- In this paragraph the authors refer to the provisions of the Law 24/2017 but in the footnote mention the provisions of the Italian Civil Code on contractual liability and tortious liability. A brief clarification on the effects of the law 24/2017 on these two concepts should be provided.
Line 94- ”in” before ”contractual liability” should be deleted
Line 119- a bibliographic reference is required
Lines 139, 171- the paragraphs should begin with small letter- for the sake of editorial consistency
Lines 208-211- the authors mention a number of 43 cases in which only physicians were involved and then mention the number of cases in different specialties. However, the sum of the numbers mentioned for each specialty is 38 and not 43- Please check!
Figure 1- should be placed after line 219. Figure 1 must be accompanied by a legend explaining what each segment represents. A title to figure 1 should be added.
Line 224- ”ex-Article 696 bis”- please also mention the Law
Figure 2- a title should be added
Table 1 is not necessary as the same findings are clearly represented in figure 3. A title should be added to figure 3.
Line 255- ”...” at the end of the text in parentheses should be deleted
Table 2, first row- ”convictions” instead of ”convintious”. A title should be added to table 2
Lines 262, 263- that text would better fit in the Discussion section
Line 286- parantheses- ”in Italy” should be deleted
Line 289, 291- ”healthcare” to be added before ”public facility”
Line 306- it is not clear at the beginning of which paragraph the authors refer
The author should check for accuracy the following paragraphs: Lines 317, 318- ”the average interval between the adverse event and the settlement of the judgement was equal to 5.3 years and the average interval between its settlement and the publication of the judgement (coinciding with its end) was 4.3 years” and Lines 194- 197- ”an average 194 period of 5.3 years between the claimed event and the start of the actual dispute, while the interval between the registration on the docket (that is its start) and the publication of the judgement is about 4.3 years”. ”Settlement of the judgement” does not have the same meaning as ”the start of the actual dispute”.
Line 352- a bibliographic reference is required
Line 361- ”claimed” instead of ”required”
Line 367- a bibliographic reference is required
Line 364- the authors mention that the informed consent has been recently introduced. They should also mention where the informed consent was recently introduced.
Reviewer 3 Report
I think that the paper involves a high interesting issue, both in an international and national point of view, regarding the new Laws. It gives an overview of the civil orientation soon after the Gelli-Bianco Law and it would be very interesting a future comparison with more recent Judgements' orientations. It must be improved in the literature review and background, because too little was said about the new 24/2017 Law and the insurance recourse from the facilities to the health care personnel. Moreover, the cited literature involved only some Authors; instead, many other authoritative commentaries (volumes, books, articles...) about Italian MedMal and "loss of chances" exist and I think they must be mentioned in such a study.
Some Tables and Figures are repetitive (Tab 1 and Fig. 3 are just the same). On the contrary some other aspects that were collected in the Excel were not discussed, as the options "defendant/facilities" or "defendant/persons".
In all Tables there should be an order: for example an alphabetic order of the specialties, or a decrescent order of the convictions or compensations.
Typos and grammar errors:
- extra-spaces in lines: 49, 325, 373, 375
- extra comma, before the verb: lines 102 and 333
- yes and no in line 163 should be written "yes" and "no"
- line 238: do not use "we", but "the Authors"
- line 331: it is better "likely" than "probable"
Reviewer 4 Report
Although I did enjoy most of this paper, I doubt about its appropriateness amd the attractiveness to readers. My major concern rests not in the paper itself (paper is well written and well organized, the research presented was well-conducted….), but in the appeal of this specific paper in this specific journal, in other words – I don’t think it It falls within the Aims and Scope.
